# Influence of Carboxymethyl Cellulose as a Thickening Agent for Glauber’s Salt-Based Low Temperature PCM

**DOI:** 10.3390/ma17102442

**Published:** 2024-05-18

**Authors:** Jay Thakkar, Sai Bhargav Annavajjala, Margaret J. Sobkowicz, Jan Kosny

**Affiliations:** 1Department of Plastics Engineering, University of Massachusetts Lowell, Lowell, MA 01854, USA; jay_thakkar@student.uml.edu (J.T.); margaret_sobkowiczkline@uml.edu (M.J.S.); 2Department of Mechanical Engineering, University of Massachusetts Lowell, Lowell, MA 01854, USA; saibhargav_annavajjala@student.uml.edu

**Keywords:** PCM, salt hydrates, Glauber’s salt, sodium sulphate decahydrate, CMC, thickening agent, cold storage, thermal energy storage

## Abstract

This work is focused on a novel, promising low temperature phase change material (PCM), based on the eutectic Glauber’s salt composition. To allow phase transition within the refrigeration range of temperatures of +5 °C to +12 °C, combined with a high repeatability of melting–freezing processes, and minimized subcooling, the application of three variants of sodium carboxymethyl cellulose (Na-CMC) with distinct molecular weights (700,000, 250,000, and 90,000) is considered. The primary objective is to optimize the stabilization of this eutectic PCM formulation, while maintaining the desired enthalpy level. Preparation methods are refined to ensure repeatability in mixing components, thereby optimizing performance and stability. Additionally, the influence of Na-CMC molecular weight on stabilization is examined through differential scanning calorimetry (DSC), T-history, and rheology tests. The PCM formulation of interest builds upon prior research in which borax, ammonium chloride, and potassium chloride were used as additives to sodium sulfate decahydrate (Glauber’s salt), prioritizing environmentally responsible materials. The results reveal that CMC with molecular weights of 250 kg/mol and 90 kg/mol effectively stabilize the PCM without phase separation issues, slowing crystallization kinetics. Conversely, CMC of 700 kg/mol proved ineffective due to the disruption of gel formation at its low gel point, hindering higher concentrations. Calculations of ionic concentration indicate higher Na ion content in PCM stabilized with 90 kg/mol CMC, suggesting increased ionic interactions and gel strength. A tradeoff is discovered between the faster crystallization in lower molecular weight CMC and the higher concentration required, which increases the amount of inert material that does not participate in the phase transition. After thermal cycling, the best formulation had a latent heat of 130 J/g with no supercooling, demonstrating excellent performance. This work advances PCM’s reliability as a thermal energy storage solution for diverse applications and highlights the complex relationship between Na-CMC molecular weight and PCM stabilization.

## 1. Introduction

Phase change materials (PCMs) have emerged as a vital technology for efficient energy storage and thermal management and today, they are considered as key elements of energy efficient technologies and energy storage methods. It is important to highlight that because of their natural abundance, thermal energy storage using PCMs is referenced in many publications as an important component of sustainable/green energy systems. Since currently used PCM technologies exhibit a high density of energy storage and display demand leveling at the phase change temperature, their enthalpy of fusion can be utilized in a variety of thermal applications, including building energy efficiency and cold-chain applications, which are the main objective of this paper. Among all PCM types, inorganic salt hydrates stand out due to their low cost, fire resistance, and ability to store and release significant amounts of thermal energy during phase transitions. These materials, formed by combining salts with water molecules, offer high heat storage capacity within a narrow temperature range, making them attractive for applications like buildings, refrigeration, solar energy systems, cold chain applications, and waste heat recovery [1,2,3,4,5,6]. 

Refrigeration technologies are considered the most effective means of improving building energy efficiency, improving battery operation in electric vehicles, and enabling the low-cost shipping of temperature sensitive products (medical/pharmaceutical goods, preserved food, etc.). They contribute significantly to a wide range of energy usage. As a result, it is critical to enhance their performance and durability [7].

Salt hydrate-based thermal storage technologies are emerging in both building sectors, as well as in cold chain applications. They possess several key advantages such as a narrow temperature range of phase transition processes, high energy density, and relatively high thermal conductivity when comparing to organic PCMs. They have great potential for enhancing thermal energy storage systems. However, challenges remain, including improving phase change kinetics, phase separation, cycling stability, and encapsulation methods [8,9]. Phase separation, a common phenomenon in phase change materials (PCMs), refers to the uneven distribution of components during phase transitions, leading to irreversibly reduced material performance. In the context of salt hydrate PCMs, phase separation can lead to lower hydrate formation or partial material participation in the phase transition process. Furthermore, phase separation can hinder their effectiveness by causing uneven energy storage and release. Maintaining high energy storage density and long-term durability, while minimizing the sub-cooling effect, are key factors enabling future market implementation and product adoption. Both PCM enthalpy and the subcooling effect can be controlled by the selection of basic chemical precursors; however, assuring long term durability is very complicated and involves both the precise usage of additives, and the development of a reliable synthesis method. 

A common strategy to mitigate the phase separation issue is the incorporation of a suitable thickening agent. Thickening or gelation agents help maintain a homogeneous mixture of salts and water molecules, promoting consistent phase transitions and physically preventing phase separation. The choice and amount of thickening agent strongly impact the stability and thermophysical properties of the PCM. Additionally, thickening agents play a pivotal role in enhancing the reliability and efficiency of salt hydrate based PCMs, ensuring long-term performance [10].

Supercooling is a key performance problem caused by deficient nucleation conditions, and that is also indirectly related to the stability of inorganic salt based PCMs. Supercooled PCM remains in a liquid state below its expected freezing point due to a delay in crystallization. This delay can occur when nucleation sites for crystallization are lacking or when the nucleation process is hindered by impurities or insufficient agitation. To mitigate supercooling, nucleating agents or seeding materials can be introduced to encourage prompt crystallization at the intended phase transition temperature. These agents provide surfaces for crystals to form, reducing the energy barrier and promoting the timely transition from liquid to solid state, enhancing the stability and reliability of the PCM’s thermal energy storage capabilities [11,12,13,14].

Glauber’s salt (sodium sulfate decahydrate, Na_2_SO_4_.10H_2_O, -SSD) is one of the longest used PCMs with wide natural availability and associated low cost. SSD is a prominent PCM among salt hydrates with a phase transition temperature of 32 °C. SSD presents both advantages and disadvantages as a PCM. One of its notable advantages is its high heat of fusion, allowing for substantial energy storage capacity during phase transitions. SSD also exhibits high thermal conductivity and a low cost, making it an attractive option for various thermal energy storage applications [15,16,17]. The drawbacks of SSD include its supercooling characteristics and phase separation challenges, which must be carefully controlled for practical applications. Supercooling can be suppressed using borax as a nucleating agent enhancing salt crystal formation. Many works have documented the development of SSD-based PCM formulations with melting temperatures around +32 °C. However, the major goal of this effort, and at the same time, the main challenge, was to develop a stable SSD-based PCM formulation of phase transition within the refrigeration range of temperatures between +5 °C and +12 °C. For this purpose, KCl and NH_4_Cl were used as melting point depressants. A mixture of these three salts at the eutectic composition has a lower melting temperature than the pure Glauber’s salt. It is noteworthy that the widely discussed supercooling problem in many typical SSD-based PCM formulations (of +32 °C melting temperature) has already been solved (suppressed to below 1 °C), by using 2–5 wt% of borax [11,16,18]. However, phase separation suppression is still a problem with a loss of latent heat over long-term thermal cycling. Numerous thickening agents have been used to suppress the phase segregation of inorganic PCMs, yielding stable latent heat over long-term thermal cycling [19,20]. Commonly used thickening agents include cellulose derivatives, such as hydroxyethyl cellulose (HEC), cellulose nanofiber (CNF), and carboxymethyl cellulose (CMC), as well as fumed silica, potassium polyacrylate, dextran sulfate sodium (DSS) and poly (sodium 4-styrenesulfonate) (PPS). Other polymeric compounds such as guar gum, xanthan gum, alginates, modified starch, sodium polyacrylate, polyacrylamide and agar are also commonly used. In addition to the major requirement of maintaining PCM’s stability, each thickening agent is associated with specific parameters/conditions which need to be carefully considered, such as acidic/basic conditions, molecular weight, associated salts, particle mesh, material mesh, heating time, mixer speed, temperature, the presence of salts, and the order in which these additives are added and combined [10,21,22]. In a prior study conducted by Akamo D. et al., the stability of solid–solid dispersion PCMs was investigated, specifically comparing the effects of polyelectrolyte-based thickeners and polymeric thickeners. Notably, when CMC was employed as a thickener in their research, it resulted in the formation of a thin and inhomogeneous mixture. Consequently, no further tests were conducted due to these unfavorable outcomes [22].

When incorporating a thickening agent, the primary objectives are to prevent the separation of components within the material, reduce changes in volume, and attain a stable shape that is impervious to water loss. The increased viscosity prevents the diffusion of salt ions and maintains structural integrity throughout repeated thermal cycles. Thickening agents serve the purpose of augmenting the thickness or viscosity of a fluid, while gelling agents create a structured solid network known as a gel [23].

Thickening agents operate through three mechanisms: non-associating, associating (physical), or chemical. In the non-associating mechanism, fluid thickens because of (i) the entanglement of higher molecular weight chains, which results in an increase in viscosity, (ii) pseudoplastic behavior, and (iii) elastic properties [24]. Associating thickeners are water-soluble or swellable polymers containing hydrophobic backbones. They can have relatively lower molecular weight when compared to non-associating thickeners. Physical gelation falls into different categories: hydrophobic interactions, hydrogen bonding, ionic interaction (using di- or trivalent counter ions), or a combination of these methods. These gels are reversible, meaning they can be broken to form a solution, and their strength can vary based on temperature or pH level. Ionic interaction gels use specific ions to link the polymer chains together. The concentration of these ions affects how strong the gel becomes because more ions mean more links between the polymers. Hydrogen bonding gels are responsive to changes in the pH and/or temperature of a solution containing the polymer. Hydrophobic interaction gels happen when certain parts of the polymer, some liking water and some not, repel each other under the right conditions, causing the gel to form. In the chemical mechanism, cross-linking occurs because of the ability of functional groups to react and form covalent bonds. Chemically cross-linked gels, however, are irreversible [25,26].

CMC is a water-soluble cellulose derivative, which is often used to stabilize inorganic salt hydrate formulations. It is formed by grafting carboxymethyl groups from the hydroxyls present on the cellulose backbone, and it is predominantly supplied as a powder and with sodium counter ions to the carboxylate groups (Na-CMC). The degree of substitution refers to the average number of hydroxyl groups replaced with carboxyl groups per glucose unit, e.g., from 0 to 3. CMC has a wide range of applications in various industries including food, pharmaceuticals, cosmetics, and personal care products, among others. One of the most significant properties of CMC is its ability to thicken and stabilize suspensions due to increased viscosity. The determination of the gel point, or the concentration at which no flow is observed, is an important starting point in implementing a particular viscosity modifier. The gel point of suspensions of hydrated salts in water with a stabilizing agent such as CMC depends on various factors such as the concentration and type of the salt, molecular weight of CMC, degree of substitution, temperature, ionic strength, and pH of the solution [27]. In the case of Na-CMC, the functional groups present in the backbone not only aid in salt dissociation, but also contribute to the entrapment of anions through the formation of hydrogen bonds. This unique feature enhances the ionic conductivity of the polymer electrolyte. A high degree of substitution or concentration of CMC will have higher sodium contents available for ionic interaction, leading to higher gel strength [28]. Hydrated salts may require a higher concentration of CMC to form a gel compared to pure water, because salts may interact with CMC and reduce the electrostatic repulsion between the polymer chains, leading to a more compact and less hydrated gel structure [29].

In this work, we conducted a comprehensive investigation into the application of three Na-CMC variants with distinct molecular weights (700 kg/mol, 250 kg/mol, and 90 kg/mol) to determine the optimal conditions for stabilizing a highly promising SSD-based eutectic PCM. The objectives were to (i) refine the preparation method, to (ii) determine a repeatable procedure for the mixing of the components, such that performance and stability are optimized, and to (iii) determine the influence of Na-CMC molecular weight on the stabilization. DSC, T-history, and rheology tests were conducted to study the material behavior with different additives. The formulation employed in this study builds upon our previous research that optimized the SSD + NH_4_Cl + KCl eutectic with sodium polyacrylate (SPA) as a thickening agent and borax as a nucleating agent [30]. It is worth noting that SPA is a fossil-derived, non-biodegradable material that can persist in the environment for extended periods, whereas CMC represents a renewable and biodegradable alternative. This choice aligns with the industry prioritization of environmentally responsible materials and processes.

## 2. Materials 

Sodium sulfate anhydrous (Na_2_SO_4_, reagent grade), ammonium chloride (NH_4_Cl, granular, laboratory-grade), potassium chloride (KCl, granular, ACS grade), and borax (Na_2_B_4_O_7_.10H_2_O) were obtained from Carolina Biological Supply Company, Burlington, NC, USA, and used as received. Sodium sulfate anhydrous was used to prepare SSD using distilled water. NH_4_Cl and KCl were used to reduce the phase transition temperature of SSD. Borax, which is also a decahydrate, was used as a nucleating agent to suppress the supercooling in the SSD-based formulation. Prior research shows 3 wt.% borax suppresses the maximum supercooling and therefore borax was added as 3 wt.% [9,29] Sodium carboxymethyl cellulose (CMC) was purchased from Sigma-Aldrich Inc, St. Louis, MO, USA. The degree of substitution for different molecular weight CMCs (700 kg/mol, 250 kg/mol, and 90 kg/mol) was 0.8, 0.89, and 0.6–0.95, respectively, as per the specification sheet of the supplier.

In this research study, a tilt test was conducted to determine the approximate gelation point of CMC in water [31]. CMC was incrementally introduced into the solution, with the concentration increasing by 1 wt.% at each step, until the point was reached where the solution no longer self-leveled upon the tilting of the vial. The gel point was considered to be at this concentration. The gel points for CMC with molar mass of 700,000, 250,000, and 90,000 g/mol were found as 2%, 8%, and 11%, respectively. Figure 1 shows the gel point proportions of CMC in water, and the tilt sequences are shown in the supporting information.

## 3. Characterization Techniques

### 3.1. Differential Scanning Calorimetry (DSC)

In this research study, the phase transition temperature (including melting and peak temperature) and the enthalpy of the PCM were investigated through differential scanning calorimetry (DSC) testing. The experiments were performed using a Discover Q20 instrument from TA Instruments (Newcastle, DE, USA). Data analysis was carried out using the TA Universal Analysis software. Each sample weighed between 5 to 10 mg and was placed in aluminum pans coated with anodic material. The heating and cooling rates were maintained at 2 °C/min, and the samples underwent heating and cooling cycles within a temperature range spanning from −35°C to 45°C. Following several thermal cycles at the same rate of 2 °C/min outside of the instrument, a DSC test was conducted on the bulk sample that had undergone the cycling process.

### 3.2. Temperature History Method (T-History)

The T-history method was initially developed to determine various thermal properties of PCMs, such as phase transition temperatures (both melting and cooling), latent heat of fusion, specific heat, and thermal conductivity, but with a focus on larger sample volumes compared to what can be achieved using differential scanning calorimetry (DSC^)^ [29,32]. This research utilized the T-history method to collect supercooling data and assess the cooling rate of the PCM. To perform these experiments, the PCM samples underwent heating and cooling processes within a water bath. Temperature measurements were obtained using a T-type thermocouple inserted into the sample, and the thermocouple was connected to a data logger (CR1000, Campbell Scientific, Logan, UT, USA), which recorded the temperature data. The experimental setup for the T-history method included the water bath, the data logger, and a laptop connected to run the PC200 software (version 4.5) for data acquisition and analysis.

### 3.3. Rheology

Rheology experiments were conducted to study the comprehensive rheological characterization of phase change material (PCM) mixed with (CMC) gels, comparing two distinct systems: CMC gels with water and CMC gels with PCM. In this study, an ARES-G2 Rheometer from TA Instruments (Newark, DE, USA) was used with 25 mm parallel plates and Peltier temperature control. The complex viscosity (η∗) storage modulus (G’), and loss modulus (G”) of pure Na-CMC solutions and optimized PCM formulations were analyzed. To evaluate the viscoelastic behavior, a frequency sweep test was performed over a range of angular frequencies from 100 to 0.1 rad/s. All tests were conducted at a controlled temperature of 25 °C. During the experiments, a strain level of 50% was applied, and there was a 30 s soak time to ensure temperature equilibrium. These experiments aimed to provide a comprehensive understanding of how the rheological properties of CMC gels are influenced when combined with PCMs.

## 4. PCM Preparation Method

Various techniques were employed to fabricate SSD-based PCM samples containing CMC. Due to issues related to agglomeration during the mixing, CMC was not incorporated as a dry powder, but rather the CMC solution was prepared first, followed by the incorporation of the salts. Prior research articles by several researchers have described agglomeration when incorporating CMC powder as a last step; therefore, in this work three different methods were used. As presented in Figure 2, the methodologies differed primarily in the point at which proper composition was attained. In Method 1, the CMC gel with final water contents was prepared, followed by adding all salt components. Methods 2 and 3 were attempted for the 700,000 g/mol CMC (CMC-700), but they were not successful. Thus, Method 1 was used for all subsequent CMC variants. 

In Method 1 (Figure 2a), the stoichiometric quantity of water required for hydrate formation (20 g total PCM sample mass) was placed in a vial and heated on a hot plate at 50 °C. Next, 2%, 8% and 11% CMC of molecular weights 700,000, 250,000, and 90,000 g/mol, respectively, (CMC-700, CMC-250, CMC-90) on a basis of the stoichiometric water amount, were subsequently added and vigorously stirred using a vortex stirrer at 800 RPM for 30 min, leading to the formation of a well-dispersed gel. Following this, anhydrous Na_2_SO_4_ was introduced, and sequentially, NH_4_Cl, KCl, and borax were added, with each step followed by stirring at 800 RPM for 20 min. The prepared sample was allowed to cool to room temperature and then refrigerated overnight to ensure its stability and uniformity.

In Method 2 (Figure 2b), a modified approach was employed to prepare PCM samples incorporating CMC-700 only. This new method was attempted because the CMC gel with final water contents was broken after the addition of anhydrous salts. Initially, a 10% excess of water was added to the stoichiometric amounts of salt hydrates, and the mixture was placed in a vial on a hot plate set to 50 °C. Subsequently, 2% CMC-700 was added on a basis of the water already in the beaker, and the resulting mixture was vigorously stirred using a vortex stirrer for 30 min, yielding a well-formed gel. Following the gel formation, anhydrous Na_2_SO_4_, NH_4_Cl, KCl, and borax were sequentially introduced to the mixture, with each addition accompanied by stirring at 800 RPM for 20 min to ensure uniform distribution. In the final step, the sample was heated at 100 °C to vaporize the excess water, to achieve the target final PCM-CMC composition. Once prepared, the sample was allowed to cool to room temperature and then refrigerated overnight.

Method 3 (Figure 2c) was introduced because the PCM sample with CMC-700 was only partially stable; after 15 cycles, low latent heat was observed and supercooling was still not mitigated. In Method 3, 2% CMC-700 solution was prepared separately. The stoichiometric amount of water was added to the vial on a hot plate at 50 °C and 1, 2, and 3% gels (on a basis of final PCM weight) from the 2% CMC gel were added. After that, anhydrous Na_2_SO_4_, NH_4_Cl, KCl, and borax were sequentially added and stirred for 20 min at each step. After the sample was prepared, it was allowed to come to room temperature and then kept in the refrigerator overnight.

## 5. Results and Discussion

### 5.1. Preparation Method Comparisons of PCM Thickened with CMC-700 

Among the samples previously tested with the addition of sodium polyacrylate (SPA, a thickening agent), borax (a nucleating agent), and KCl with NH_4_Cl (depressants for melting temperature), the proportion of borax (3%) along with NH_4_Cl and KCl together (12%) turned out to be stable with negligible supercooling and a stable phase transition temperature. We prepared the same eutectic SSD sample with 3% borax, using CMC with different molecular weights as the thickening agent following the three different methods described earlier. The compositions of the samples prepared using CMC are outlined in Table 1. The total composition of KCl, NH_4_Cl, and borax remained consistent at 4.28%, 7.72%, and 3%, respectively, on a basis of SSD, for all of the samples. The composition of SSD plus all salt additives was adjusted based on the amount of CMC added to each sample (Table 1). The nomenclature of the samples is given by the weight% of CMC added in the stoichiometric amount of water and its molecular weight. For example, sample CMC2-700 has 2 wt.% 700 kg/mol CMC in the stoichiometric amount of water.

Samples prepared with Method 1, where gel was prepared first with the stoichiometric amount of water, resulted in a broken gel as the anhydrous Na_2_SO_4_ was added. After the addition of NH_4_Cl and KCl, it completely broke and anhydrous salt settled at the bottom, with water on the top surface; consequently, CMC was not able to maintain the gel. This is likely because of the screening effect of the charges of CMC and other salts. After refrigerating overnight at −20 °C, it did not freeze completely, with water still on the top surface. DSC was not conducted because of the water on the top surface.

The sample prepared with Method 2, where 10 wt.% extra water was added and subsequently evaporated, appeared to have no visible phase segregation. DSC testing was conducted to study the phase transition behavior of that PCM. The onset temperature was considered as the melting point and obtained by taking the tangent of the curves. Figure 3a,b shows the DSC plot and supercooling data up to 15 thermal cycles for the sample with 2 wt.% of CMC. Cycle 1 showed two merged peaks that elongated the melting range, showing that the melting of this sample was slow. It was found that, up to five cycles, there was no phase segregation and the latent heat increased; however, after the 10th cycle, a second peak around 30–33 ℃ emerged, corresponding to the SSD melting peak, and grew larger after the 15th cycle. Table 2 shows the phase transition data and enthalpy for sample CMC2-700. Latent heat for melting reduced after the 10th cycle from 105 J/g to 77 J/g, which is attributed to the phase separation. Supercooling was reduced up to the 10th cycle (5℃); however, it increased after the 15th cycle (8 ℃).

During the next step, Method 3 was utilized. However, when preparing a separate CMC gel and then adding it to the eutectic formulation at 1%, 2%, and 3% by weight, we encountered a phase separation with water accumulating on the top surface, even after overnight freezing. Figure 4a,b illustrate the melted and solidified PCM, respectively. The melted PCM showed salt at the bottom and water separated at the top. Consequently, DSC analysis was not possible due to this phase segregation. Moreover, the crystallization process (resulting in partial freezing) was initiated at the relatively higher temperatures of −8 °C, −5 °C, and −7 °C for the three samples with 1%, 2%, and 3% CMC, respectively. These observations indicated a considerable degree of supercooling of 18 °C, 15 °C, and 17 °C for the respective CMC-containing samples. In conclusion, the use of Method 3 led to phase separation issues and the supercooling of the PCM, hindering the desired solidification behavior.

### 5.2. Preparation of PCM with CMC-250 and CMC-90

To investigate the influence of lower molecular weight CMC on the stability of SSD-based PCM, we conducted further testing using CMC-250 and CMC-90. The main objectives of these tests were to assess the effects of lower molecular weight CMC on reducing phase segregation and improving overall stability, and to gain insights into the kinetics of the phase change process by studying the cooling rate. By understanding the impact of different CMC variants on phase segregation and cooling rates, we can make more informed decisions and enhance the overall efficiency and reliability of the PCM system for a specific application.

As per the gel point determination and the methodology development described earlier, we proceeded to prepare samples using Method 1, which did not involve any extra water. This step was crucial to assess whether the CMC gel would break, similar to what was observed with the higher molecular weight CMC-700.

For CMC-250 and CMC-90, we prepared one sample with the gel point concentration of CMC and the stoichiometric amount of water for the entire sample (8 wt.% for CMC-250 and 11 wt.% for CMC-90), and another sample with a lower CMC concentration matching that of the gel point for CMC-700 (e.g., 2%). Table 1 shows the composition of samples prepared using Method 1.

PCM sample CMC2-250 showed no visible phase segregation. The DSC test was conducted for this sample directly after the preparation and after five thermal cycles. Figure 5a shows the DSC plot for sample CMC2-250 up to five cycles. Cycle one and cycle five had no phase segregation; however, the latent heat was still low, 67 J/g and 66 J/g, respectively. Sample CMC2-90 also showed no visible phase segregation after solidification. However, DSC testing showed phase separation with the second melting peak present at 32℃ (the melting temperature of SSD). Figure 5b shows the DSC plot for CMC2-90 up to five cycles. After thermal cycling, cycle five shows no phase separation. Latent heats for cycle one and cycle five were 88 J/g and 77 J/g. After thermal cycles, both samples had visible segregation with anhydrous salt at the bottom and salt solution on the top. This means some of the salt hydrate material is not participating in the phase transition process, which is reflected in the low latent heat. Also, supercooling was observed for both samples. Figure 6 shows the thermal data for samples up to five cycles.

The other two samples were prepared by adding the gel point proportions of CMC-250 and CMC-90 (8% and 11%, respectively) to the stoichiometric amount of water required to prepare SSD, followed by the addition of anhydrous Na_2_SO_4_, NH4Cl, KCl, and borax. Sample CMC11-90 was stable from the first cycle with a latent heat of 125 J/g and no supercooling. Figure 7 shows the DSC plot for sample CMC11-90 and CMC8-250 up to 15 cycles. The latent heat after 15 cycles was 130 J/g which indicates the stability of the material with thermal cycling. There was also no supercooling after 15 thermal cycles. Table 3 shows the data for sample CMC11-90 and CMC8-250 up to 15 cycles. Sample CMC8-250 also had no phase segregation and latent heat was 102 J/g after the first cycle with supercooling of 3.4 °C. After 15 cycles latent heat was 108 J/g with supercooling below 1 °C. The latent heat for CMC11-90 was higher than CMC8-250 after 15 cycles, and supercooling was lower than CMC8-250. The CMC-90 can likely disperse more effectively within the PCM, potentially creating a more uniform and homogeneous mixture. This in turn may lead to better energy storage and release during phase transition. It also has more mobility, which results in efficient heat transfer during phase transitions. Larger structures and a tighter network in PCM with higher molecular weight (CMC8-250 and CMC2-700) can lead to reduced mobility, which leads to a slower response to temperature changes and may not be as effective at nucleating or promoting crystallization [30].

### 5.3. T-History Analysis for Supercooling

A crystallization study was conducted to understand the effect of different molecular weight CMC at the gel point proportions used in the sample. Time taken to reach 0 °C from 50 °C was measured by using data loggers in the T-history method setup. It was found that with CMC-700, crystallization took 50 min (1 °C/min), whereas it took 88 min (1.76 °C/min) for CMC-250 and 135 min (2.7 °C/min) for CMC-90. This clearly shows that lowering the molecular weight slows down the cooling rate. This may be due to the higher CMC contents needed to gel the PCM, which reduces the thermal conductivity. The initiation of crystallization was also different with varying molecular weights of CMC. The freezing temperature of samples with CMC-700 was 5.2 °C, which means 5 °C of supercooling, whereas for CMC-250, the freezing point was 8.2 °C, having 3.4 °C of supercooling. The sample with CMC-90 had a freezing point of 11.2 °C, which means no supercooling. Figure 8 shows the freezing point and time plot of three samples with CMC of different molecular weights when added at the gel point proportion. Table 4 shows the thermal data for samples with different CMCs.

### 5.4. Rheology Analysis

Frequency sweeps were conducted for CMC gels in water for CMC-700, CMC-250, and CMC-90 with concentrations 2, 8, and 11%, respectively. Figure 9 shows the plots of G’, G”, and complex viscosity for CMC gels. It was found that CMC-700 was a gel with storage modulus higher than loss modulus, showing solid-like behavior over the whole frequency range. CMC-250 showed a crossover point from liquid-like to solid-like at about 50 rad/s, and CMC-90 showed liquid-like behavior. Low viscosity was observed with the lowest molecular weight (CMC-90), whereas CMC-250 and CMC-700 had almost the same viscosity, but all three samples thinned with increasing frequency. 

Frequency sweeps were also conducted for samples with PCM to study the viscosity change with and without PCM. Figure 10 shows the complex viscosity profile for samples with and without PCM. PCM samples with different CMC molecular weights had almost the same viscosity at higher frequency. All PCM formulations had about an order of magnitude higher viscosity at low frequency compared to the pure gels. When added to PCM, CMC-90 showed the most stability, which may be because of more ion mobility in the looser gel compared to the higher molecular weight CMC. 

The proportion of CMC-250 in water was varied (3, 4, and 5%) to achieve the low viscosity profile of CMC-90 (11%), in order to test whether the good performance of PCM with CMC-90 could be replicated by matching viscosity alone. Figure 11a shows that CMC-250 with 4% and 5% had viscosity near to that of CMC-90 (11%). PCM samples were prepared with CMC-250 (4% and 5%) holding the proportion of borax, NH_4_Cl, and KCl constant at 15%. Table 1 shows the concentration of the materials for sample CMC4-250 and CMC5-250. Both PCM samples with lower CMC-250 contents did not display gel-like behavior. The DSC analysis of the melting behavior for the first cycle is shown in Figure 11b. Latent heat was below 100 J/g for both samples and a wider melting range (6–14 °C) was observed. After the first cycle, PCM samples showed visible phase segregation with anhydrous salt settled at the bottom and water on the top surface. 

Another factor that must be considered in the stabilization of PCMs using CMC is the ionic concentration in the solution. While higher molecular weight leads to more entanglement at low concentrations, this also means there are fewer ions in the higher molecular weight CMC samples. 

In order to assess the impact of ionic concentration, the moles of sodium in CMC added to PCM at gel concentrations were calculated based on the degree of substitution (DS) provided by the supplier. A higher concentration of ions results in enhanced ionic interactions, thereby reinforcing the gel structure of the PCM. As per the specification sheet provided by the supplier for CMC, the DS of CMC-700, -250, and -90 were 0.8, 0.89, and 0.6–0.95, and the amount of CMC used in samples was 0.4, 1.6, and 2.2 g for CMC-700, CMC-250, and CMC-90, respectively. The moles of sodium were higher when CMC-90 was used, which aligns well with our DSC results where CMC11-90 performed better than CMC2-700 and CMC8-250 with high latent heat, negligible supercooling, and thermal stability. Because of the increased sodium ion concentration, interaction in the PCM sample increases, leading to ionic strength, which also increases the gel strength. Below is the ionic concentration calculation shown for CMC-700. Table 5 shows the moles and mass of sodium used in PCM samples from CMC.
CMC (700,000) DS = 0.80
Moles of CMC = 0.4/700,000 = 5.7 × 10^−7^
Moles of sodium (Na) = (5.7 × 10^−7^ moles) × 0.8 = 4.56 × 10^−7^
Mass of sodium (Na) = moles × molar mass = 4.56 × 10^−7^ × 23 = 1.05 × 10^−5^

## 6. Conclusions

This study was successful in finding a stable low temperature Glauber’s salt based PCM formulation with CMC and it also demonstrated the complex interactions of CMC when used as a thickener for PCMs. Adding CMC directly in the stoichiometric amount of water proved to be the most effective method for incorporating CMC without agglomeration. Using the extra water principle and preparing CMC gel separately did not prevent the agglomeration. This study showed that the effectiveness of CMC varies with its molecular weight as well as the concentration added. In our experiments, CMC-700 proved ineffective because the salts added disrupted the gel formation at its low gel point of 2 wt%, but higher concentrations were not possible due to difficulties with dissolving the less mobile polymer chains. However, CMC with molecular weights of 250,000 g/mol and 90,000 g/mol successfully stabilized the PCM without any phase separation issues, while also slowing the crystallization kinetics. It is worth noting that higher molecular weight CMC promotes faster crystallization by restricting molecular mobility, whereas it seems that molecular mobility leads to the stability in PCM with lower molecular weight. These effects highlight the potential for fine-tuning molecular characteristics to control crystallization dynamics in PCM formulation. The calculations of the ionic concentration for each CMC type showed a one to two orders of magnitude higher amount of Na ions in CMC11-90 compared to CMC8-250 and CMC2-700, which we speculate leads to higher ionic interactions and gel strength. These results show that, while high molecular weight can prevent the diffusion of ions that leads to phase separation, the low mass added results in insufficient ion concentration to stabilize the gel with PCM present. After subjecting sample CMC11-90 to 15 thermal cycles, we observed a significant latent heat of 130 J/g with no supercooling, indicating its excellent performance. The drawback of using the lower molecular weight CMC is that more inert material is present that does not participate in the phase transition. Thus, a tradeoff exists in the selection of the CMC molecular weight for stabilizing these formulations. This advancement enhances PCM’s performance and renders it a dependable and efficient thermal energy storage solution suitable for a wide range of applications.

## Figures and Tables

**Figure 1 materials-17-02442-f001:**
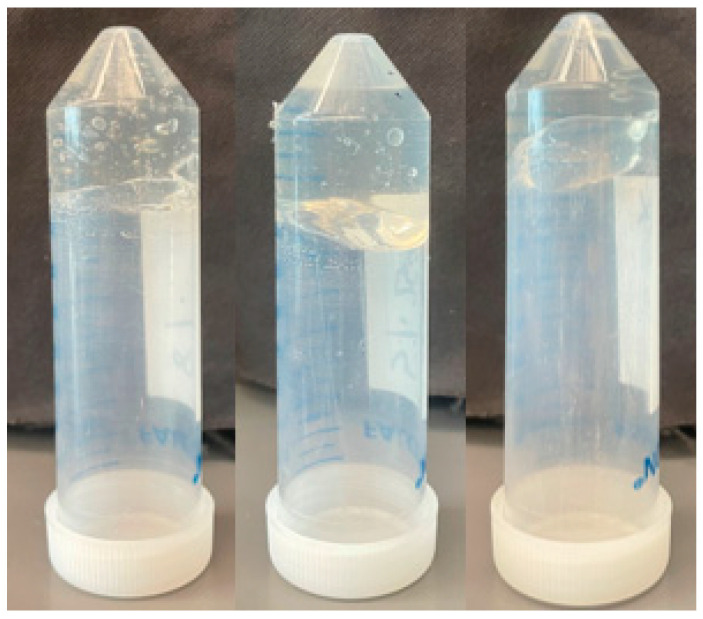
CMC gels with molecular weights 700,000, 250,000, and 90,000 g/mol at 2, 8, and 11 wt% concentrations (left to right).

**Figure 2 materials-17-02442-f002:**
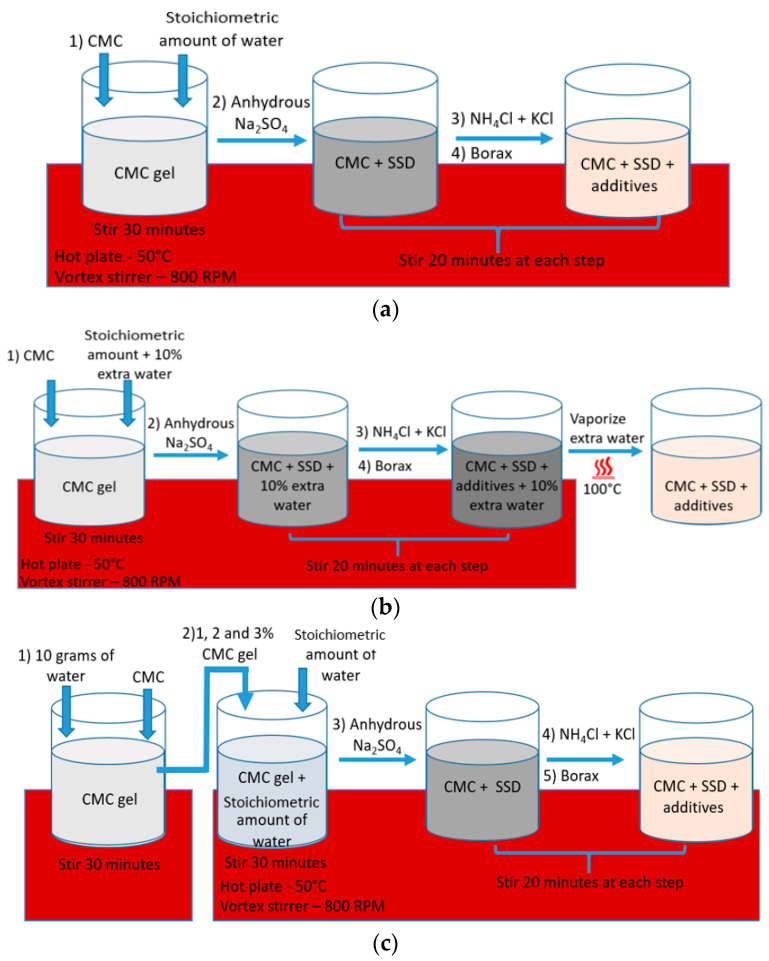
PCM sample preparation with (**a**) PCM salts added to whole gel, Method 1; (**b**) extra water, Method 2; and (**c**) pre-gel, Method 3.

**Figure 3 materials-17-02442-f003:**
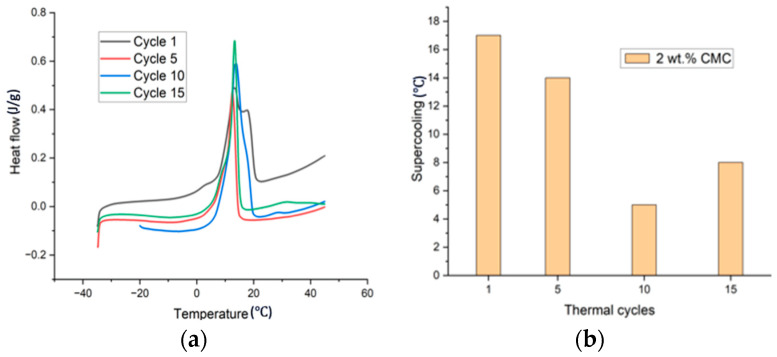
(**a**) DSC plot and (**b**) supercooling data up to 15 cycles with 2 wt.% CMC2-700.

**Figure 4 materials-17-02442-f004:**
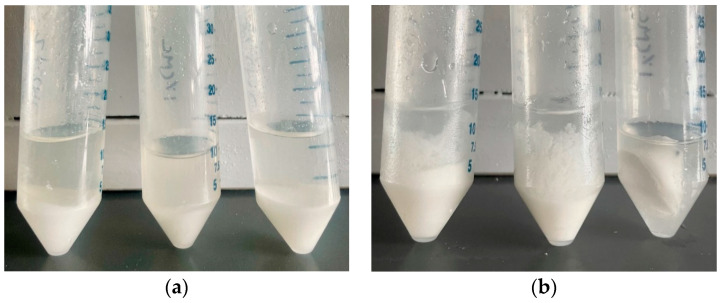
PCM samples with Method 3 of 1, 2, and 3 wt.% CMC gel: (**a**) melted and (**b**) solidified.

**Figure 5 materials-17-02442-f005:**
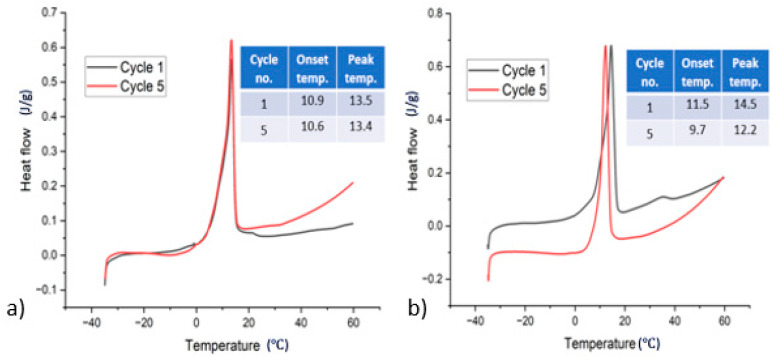
DSC plots for samples with (**a**) CMC2-250 and (**b**) CMC2-90.

**Figure 6 materials-17-02442-f006:**
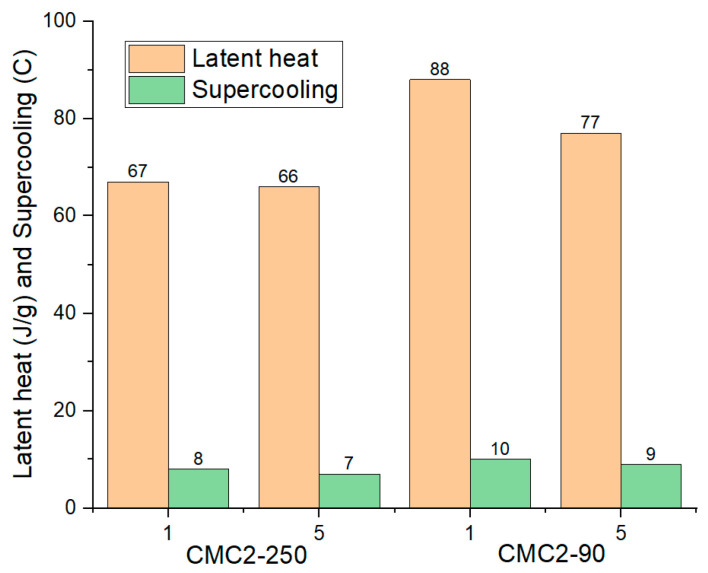
Thermal data for samples CMC2-250 and CMC2-90 up to cycle 5.

**Figure 7 materials-17-02442-f007:**
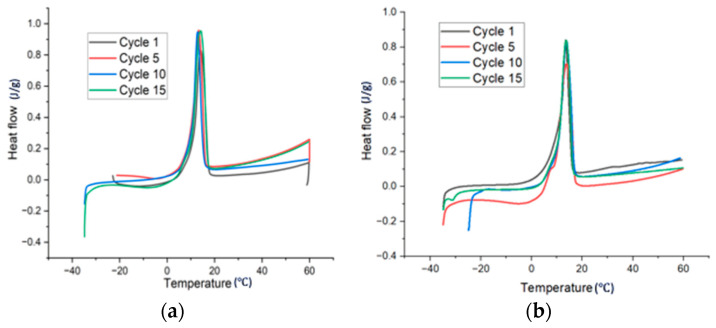
DSC plots for samples (**a**) CMC11-90 and (**b**) CMC8-250 up to 15 cycles.

**Figure 8 materials-17-02442-f008:**
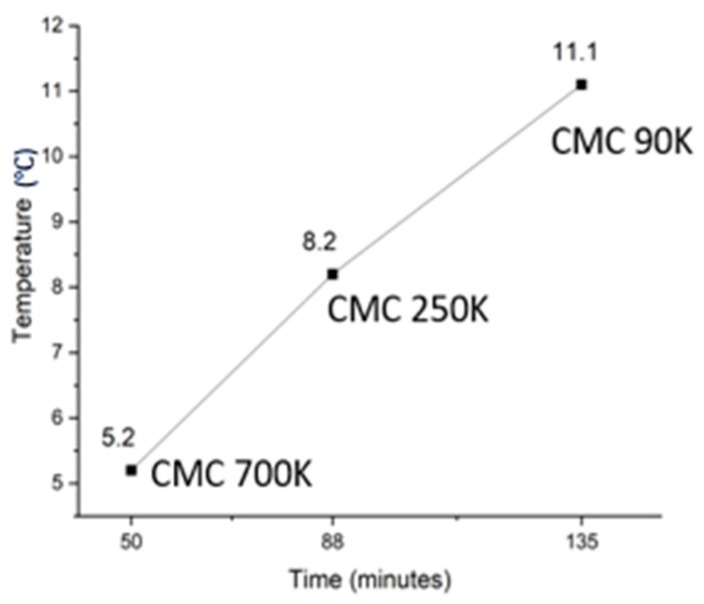
Time vs freezing temperature plot for different molecular weights of CMC taken from the T-history measurements.

**Figure 9 materials-17-02442-f009:**
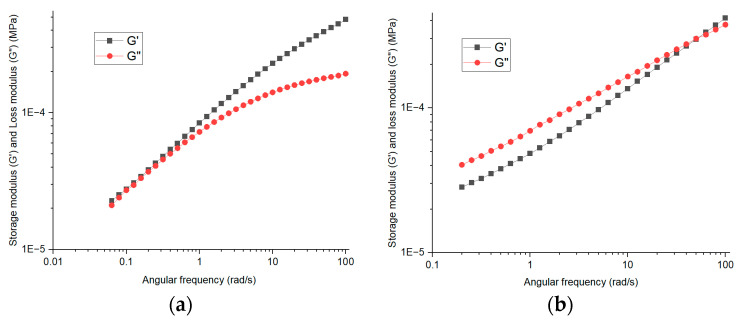
G’ and η* plots for CMC gel with water, (**a**) CMC-700 2%, (**b**) CMC-250 8%, (**c**) CMC-90 11%, and (**d**) complex viscosity for all three CMC gels%.

**Figure 10 materials-17-02442-f010:**
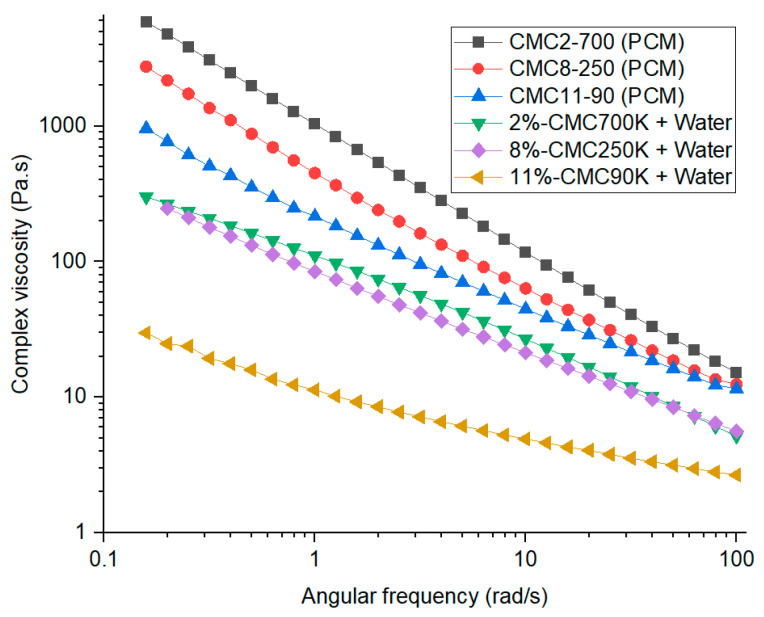
Complex viscosity plots for various molecular weight CMC gels at gel point w/ and w/o PCM.

**Figure 11 materials-17-02442-f011:**
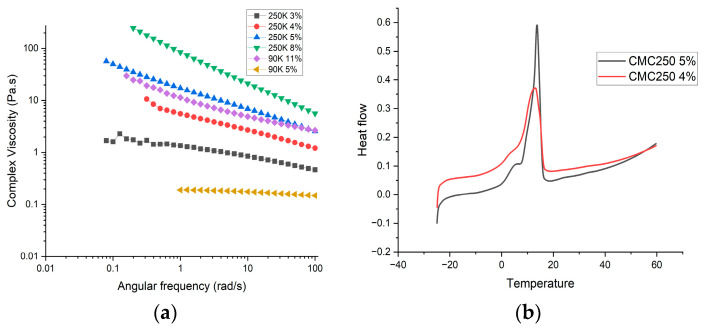
(**a**) Complex viscosity plots for CMC gels (CMC250 3, 4, 5, and 8% and CM90 5 and 11% with water). (**b**) DSC plots for PCM samples with CMC4-250 and CMC5-250.

**Table 1 materials-17-02442-t001:** Composition of PCM samples prepared with CMC.

Method	Sample No.	SSD + KCl + NH_4_Cl + Borax (%)	CMC Powder (%)	CMC Gel (2 wt.%)
1	CMC2-700	98	2	-
2	CMC2-700E	98	2	-
3	CMC1-700	99	-	1
3	CMC2-700	98	-	2
3	CMC3-700	97	-	3
1	CMC2-250	98	2	-
1	CMC2-90	98	2	-
1	CMC8-250	96.32	3.68	-
1	CMC11-90	95.07	4.93	-
1	CMC4-250	98.16	1.84	-
1	CMC5-250	97.70	2.30	-

**Table 2 materials-17-02442-t002:** Phase transition temperature and latent heat data for CMC2-700 up to 15 cycles.

Cycle No.	Melt Temp. (°C)	Peak Temp. (°C)	Enthalpy (J/g)
1	9.8	12.6	80
5	9.5	13.2	90
10	10.5	13.8	105
15	11.3	13.2	77

**Table 3 materials-17-02442-t003:** Thermal data for samples CMC11-90 and CMC8-250 up to 15 cycles.

Sample No.	Cycle No.	Onset Temp.	Peak Temp.	Latent Heat
CMC11-90	1	10.75	14.19	125
	5	10.32	13.32	124
	10	10.28	12.90	128
	15	10.15	14.12	130
CMC8-250	1	11.6	13.4	102
	5	10.3	13.7	107
	10	9.1	13.9	115
	15	10.2	13.8	108

**Table 4 materials-17-02442-t004:** Thermal data for comparing samples with varying molecular weights of CMC at gel point proportions.

Sample	Onset Temp. (°C)	Peak Temp. (°C)	Freezing Temp. (°C)	Supercooling (°C)	Time to Reach 0 °C (min)
CMC2-700	10.2	13.8	5.2	5	50
CMC8-250	11.6	13.4	8.2	3.4	88
CMC11-90	10.1	14.2	11.2	0	135

**Table 5 materials-17-02442-t005:** DS, moles of sodium, and mass of sodium in PCM samples.

Na-CMC mw (g/mol)	DS	Na-CMC Mass in Samples (g)	Moles of Sodium (mol)	Mass of Sodium (g)
90,000	0.60 0.95	2.2	1.44 × 10^−5^ 2.28 × 10^−5^	3.30 × 10^−4^ 5.23 × 10^−4^
250,000	0.89	1.6	5.70 × 10^−6^	1.31 × 10^−4^
700,000	0.80	0.4	4.56 × 10^−7^	1.05 × 10^−5^

## Data Availability

Data is available upon request.

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
