# Peer review of "Influence of Carboxymethyl Cellulose as a Thickening Agent for Glauber’s Salt-Based Low Temperature PCM"

_materials, 2024, doi:10.3390/ma17102442_

Round 1
Reviewer 1 Report
Comments and Suggestions for Authors
In this manuscript, Kosny et al. report a comprehensive investigation into the application of three Na-CMC variants with distinct molecular weights (700 kg/mol, 250 kg/mol, and 90 kg/mol) to determine the optimal conditions for stabilizing a highly promising SSD based eutectic PCM. The following comments need to be addressed.
1. The naming convention of samples is quite intricate. Could you provide some insight into what the numbers (1, 2, 3, 8, 11...) behind CMC stand for? It’s hard to connect the names with the corresponding compositions shown in Table 1.
2. (Table 1) What are the methods for rows 4 and 5?
3. (Line 275-276) Why are there two peaks in the black plot (Cycle 1)?
“after the 10th cycle, a second peak emerged” Please indicate which peak you are talking about, the small bump around 30 C?
4. (Table 2) Based on the plots in Figure 3, it looks like the melt temperatures of all cycles start at around 5 C. How did you determine the onset temperatures? Based on tangent?
5. What does the y-axis of Figure 3b stand for, any unit? How did you get the “supercooling” data for Figure 3b? Add unit to y-axis of Figure 8.
6. Why do the baselines in Figures 3, 5, and 7 shift a lot? In Figure 7, it looks like different cycles have different start temperatures.
7. The data in all the tables (table 2, 3, 4…) and column graphs (Figure 3b, 6…) are shown without any standard deviations and error bars. For example, DSC is quite sensitive and even sample preparation can lead to some deviations. Have you replicated the DSC measurement of each sample several times?
8. The name for the same sample should be consistent. For example, CMC-250 and CMC 250K both appear in the manuscript.
9. (Line 416) “When added to PCM, CMC-90 showed the most stability” Do you mean a lower slope? If so, it's hard to compare the slope with a log scale y-axis.
10. The authors claim “This study was successful in finding a stable low temperature Glauber’s salt based PCM formulation with CMC and it also demonstrated the complex interactions of CMC when used as a thickener for PCMs.” How does the performance of CMC compare with other thickeners?
11. Information about instruments and corresponding experimental details are missing.
Comments on the Quality of English LanguageModerate editing is needed
Author Response
Reviewer 1
Comments and Suggestions for Authors
In this manuscript, Kosny et al. report a comprehensive investigation into the application of three Na-CMC variants with distinct molecular weights (700 kg/mol, 250 kg/mol, and 90 kg/mol) to determine the optimal conditions for stabilizing a highly promising SSD based eutectic PCM. The following comments need to be addressed.
- The naming convention of samples is quite intricate. Could you provide some insight into what the numbers (1, 2, 3, 8, 11...) behind CMC stand for? It’s hard to connect the names with the corresponding compositions shown in Table 1.
- Added a line addressing that.
- (Table 1) What are the methods for rows 4 and 5?
- Added in Table 1
- (Table 2) Based on the plots in Figure 3, it looks like the melt temperatures of all cycles start at around 5 C. How did you determine the onset temperatures? Based on tangent?
- Explained in that paragraph. Yes, onset temperatures are based on tangent method on curves.
- What does the y-axis of Figure 3b stand for, any unit? How did you get the “supercooling” data for Figure 3b? Add unit to y-axis of Figure 8.
- Figures have been edited.
- Why do the baselines in Figures 3, 5, and 7 shift a lot? In Figure 7, it looks like different cycles have different start temperatures.
- Yes, few samples were started at different cooling temperature. It didn’t make any difference in results because of DSC temperatures much lower than cooling temperatures.
- The data in all the tables (table 2, 3, 4…) and column graphs (Figure 3b, 6…) are shown without any standard deviations and error bars. For example, DSC is quite sensitive and even sample preparation can lead to some deviations. Have you replicated the DSC measurement of each sample several times?
- Yes, all samples were prepared 2 times to see how repeatable the results are. Those results were aligning well with the published results.
- The name for the same sample should be consistent. For example, CMC-250 and CMC 250K both appear in the manuscript.
- Rheology sample names have been revised to eliminate the “K”
- (Line 416) “When added to PCM, CMC-90 showed the most stability” Do you mean a lower slope? If so, it's hard to compare the slope with a log scale y-axis.
- Yes because of more ion mobility in the looser gel compared to higher molecular weight CMC. Also this conclusion is supported by no phase segregation and higher latent heat.
- The authors claim “This study was successful in finding a stable low temperature Glauber’s salt based PCM formulation with CMC and it also demonstrated the complex interactions of CMC when used as a thickener for PCMs.” How does the performance of CMC compare with other thickeners?
- Initially this formulation was used in our research with sodium polyacrylate (PAAS) which will be used as a reference in this paper. The majority of the PCM manufacturing companies and journal articles focus on CMC; however, the formulations published are not replicable in terms of preparation method and thermal properties. This article focused on preparation methods with CMC and how molecular weight can impact the overall PCM properties.
- Information about instruments and corresponding experimental details are missing.
- Characterization technique section added.
Reviewer 2 Report
Comments and Suggestions for Authors
This manuscript aimed to show the influence of sodium carboxymethyl cellulose (CMC) with different molecular weights on the stabilization of phase change materials. The authors were able to find a good CMC candidate and demonstrate the trade-offs when using low molecular weight CMC as a thickening agent. Overall, this manuscript provides sufficient data to support its conclusions and is of interest to the research community. I only have some minor comments.
1. The manuscript spent a considerable amount of time discussing three different preparation methods and eventually chose one of them. However, I do not see the authors give enough background for the needs to consider these three methods. In addition, the authors did not provide useful conclusions from comparing the results of these three methods. Thus, I think the authors need to either move the description and comparison of the three methods to supplementary materials (only show the best one) or add the missing background and insights to legitimate the discussion of different methods.
3. In line 388, what is ‘CMCsI’?
2. In Figure 8, I do not see the need to use lines to connect the points unless the authors want to fit the data. I think just showing the points is good enough.
3. In line 417, the authors mentioned that ‘CMC-90 showed the most stability’. I was wondering if the authors could explain more about how they reached this conclusion based on the data.
4. In conclusion, I was wondering if the authors could discuss more about the insights for finding good CMC for the purpose of thickening agents based on their current results.
Author Response
Reviewer 2
This manuscript aimed to show the influence of sodium carboxymethyl cellulose (CMC) with different molecular weights on the stabilization of phase change materials. The authors were able to find a good CMC candidate and demonstrate the trade-offs when using low molecular weight CMC as a thickening agent. Overall, this manuscript provides sufficient data to support its conclusions and is of interest to the research community. I only have some minor comments.
- The manuscript spent a considerable amount of time discussing three different preparation methods and eventually chose one of them. However, I do not see the authors give enough background for the needs to consider these three methods. In addition, the authors did not provide useful conclusions from comparing the results of these three methods. Thus, I think the authors need to either move the description and comparison of the three methods to supplementary materials (only show the best one) or add the missing background and insights to legitimate the discussion of different methods.
- It is already in the description: “Due to issues related to agglomeration during the mixing, CMC was not incorporated as a dry powder, but rather the CMC solution was prepared first, followed by incorporation of the salts. Prior research articles by several researchers have mentioned about incorporating CMC powder as a last step which creates agglomeration because of which in this work, three different methods were used.”
- Adding CMC directly in the stoichiometric amount of water proved to be the correct method for incorporating CMC without agglomeration showing its efficiency as a thickening agent. Extra water principle and preparing CMC gel separately didn’t prevent the agglomeration.
- In line 388, what is ‘CMCsI’?
- Writing error.
- In Figure 8, I do not see the need to use lines to connect the points unless the authors want to fit the data. I think just showing the points is good enough.
- It was just done to show the incremental behavior of freezing temperature. A sentence was added to the caption to indicate this is a line to guide the eye.
- In line 417, the authors mentioned that ‘CMC-90 showed the most stability’. I was wondering if the authors could explain more about how they reached this conclusion based on the data.
- More ion mobility with lower viscosity and less entanglement with lower molecular weight
- High latent heat compared to higher molecular weight.
- Efficiently mitigating phase segregation
- In conclusion, I was wondering if the authors could discuss more about the insights for finding good CMC for the purpose of thickening agents based on their current results.
- Some modifications were made to the conclusion, see the revised text here:
- This study was successful in finding a stable low temperature Glauber’s salt based PCM formulation with CMC and it also demonstrated the complex interactions of CMC when used as a thickener for PCMs. Adding CMC directly in the stoichiometric amount of water proved to be the most effective method for incorporating CMC without agglomeration. Using the extra water principle and preparing CMC gel separately did not prevent the agglomeration. This study showed that the effectiveness of CMC varies with its molecular weight as well as the concentration added. In our experiments, CMC-700 proved ineffective because the salts added disrupted the gel formation at its low gel point of 2 wt%, but higher concentrations were not possible due to difficulties with dissolving the less mobile polymer chains. However, CMC with molecular weights of 250,000 g/mol and 90,000 g/mol successfully stabilized the PCM without any phase separation issues, while also slowing the crystallization kinetics. It is worth noting that higher molecular weight CMC promotes faster crystallization by restricting molecular mobility, whereas it seems that molecular mobility leads to the stability in PCM with lower molecular weight. These effects highlight the potential for fine-tuning molecular characteristics to control crystallization dynamics in PCM formulation. Calculations of the ionic concentration for each CMC type showed one to two orders of magnitude higher amount of Na ions in CMC11-90 compared to CMC8-250 and CMC2-700, which we speculate leads to higher ionic interactions and gel strength. These results show that while high molecular weight can prevent the diffusion of ions that leads to phase separation, the low mass added results in insufficient ion concentration to stabilize the gel with PCM present. After subjecting sample CMC11-90 to 15 thermal cycles, we observed a significant latent heat of 130 J/g with no supercooling, indicating its excellent performance. The drawback of using the lower molecular weight CMC is that more inert material is present that does not participate in the phase transition. Thus, a tradeoff exists in selection of the CMC molecular weight for stabilizing these formulations. This advancement enhances PCM's performance and renders it a dependable and efficient thermal energy storage solution suitable for a wide range of applications.
Reviewer 3 Report
Comments and Suggestions for Authors
In this work, the application of three variants of sodium carboxymethyl cellulose (Na-CMC) with distinct molecular weights (700,000, 250,000, and 90,000) for optimizing the stabilization of a promising Glauber's salt-based phase change material (PCM) with a eutectic composition has been investigated. The researchers are successful finding out a stable low temperature formulation with CMC as a thickening agent. The main focus of this research is to (i) refine the preparation method, to (ii) determine a repeatable procedure for mixing of the components, such that performance and stability are optimized, and to (iii) determine the influence of Na-CMC molecular weight on the stabilization.
The strengths of this manuscripts are as follows.
o The objectives of the research that the researchers have outlined in the manuscript to develop a stable formulation using CMC as a thickening agent in Glauber’s salt based low temperature PCM are achieved successfully with the design of the experiments, method development, and instrumental analysis such as DSC, T – history, and rheology.
o The article is logical and well organized. No issue understanding the essence of the research.
The limitation of this manuscripts is as follows.
o In my opinion, the uniqueness of this research is not cutting-edge. It is well known that the CMC is widely used as a thickening agent. It is the formulation that the researchers have optimized using CMC with three different molecular weights to make it stable for the Glauber’s salt based low temperature PCM.
Here is a suggestion for the authors to revise – It would be helpful for the readers if the figure captions specifically the rheology plots (Figure 9-11) are illustrated furthermore with the major findings.
Author Response
Reviewer 3
- In this work, the application of three variants of sodium carboxymethyl cellulose (Na-CMC) with distinct molecular weights (700,000, 250,000, and 90,000) for optimizing the stabilization of a promising Glauber's salt-based phase change material (PCM) with a eutectic composition has been investigated. The researchers are successful finding out a stable low temperature formulation with CMC as a thickening agent. The main focus of this research is to (i) refine the preparation method, to (ii) determine a repeatable procedure for mixing of the components, such that performance and stability are optimized, and to (iii) determine the influence of Na-CMC molecular weight on the stabilization.
The strengths of this manuscripts are as follows.
- The objectives of the research that the researchers have outlined in the manuscript to develop a stable formulation using CMC as a thickening agent in Glauber’s salt based low temperature PCM are achieved successfully with the design of the experiments, method development, and instrumental analysis such as DSC, T – history, and rheology.
- The article is logical and well organized. No issue understanding the essence of the research.
- The limitation of this manuscripts is as follows.
In my opinion, the uniqueness of this research is not cutting-edge. It is well known that the CMC is widely used as a thickening agent. It is the formulation that the researchers have optimized using CMC with three different molecular weights to make it stable for the Glauber’s salt based low temperature PCM.
- We do agree that CMC has been used widely as a thickening agent. However, emphasis in this work is on preparation method and molecular weight in order to more closely study the effects of agglomeration, clustering and viscosity.
Here is a suggestion for the authors to revise – It would be helpful for the readers if the figure captions specifically the rheology plots (Figure 9-11) are illustrated furthermore with the major findings.
- Captions have been revised, thank you for the suggestion.
Round 2
Reviewer 1 Report
Comments and Suggestions for Authors
Accept in present form
Comments on the Quality of English LanguageMinor editing
Reviewer 3 Report
Comments and Suggestions for Authors
The authors have modified the manuscript as per the suggestions. It is now sufficiently improved to publish in Materials.